# Evaluation of a Triage Checklist for Mild COVID-19 Outpatients in Predicting Subsequent Emergency Department Visits and Hospitalization during the Isolation Period: A Single-Center Retrospective Study

**DOI:** 10.3390/jcm11185444

**Published:** 2022-09-16

**Authors:** Yasuhiro Tanaka, Kazuko Yamamoto, Shimpei Morimoto, Takeshi Nabeshima, Kayoko Matsushima, Hiroshi Ishimoto, Nobuyuki Ashizawa, Tatsuro Hirayama, Kazuaki Takeda, Hiroshi Gyotoku, Naoki Iwanaga, Shinnosuke Takemoto, Susumu Fukahori, Takahiro Takazono, Hiroyuki Yamaguchi, Takashi Kido, Noriho Sakamoto, Naoki Hosogaya, Shogo Akabame, Takashi Sugimoto, Hirotomo Yamanashi, Kosuke Matsui, Mai Izumida, Ayumi Fujita, Masato Tashiro, Takeshi Tanaka, Koya Ariyoshi, Akitsugu Furumoto, Kouichi Morita, Koichi Izumikawa, Katsunori Yanagihara, Hiroshi Mukae

**Affiliations:** 1Department of Respiratory Medicine, Nagasaki University Graduate School of Biomedical Sciences, Nagasaki 852-8501, Japan; 2Department of Respiratory Medicine, Nagasaki University Hospital, Nagasaki 852-8501, Japan; 3Clinical Research Center, Nagasaki University Hospital, Nagasaki 852-8102, Japan; 4Department of Virology, Institute of Tropical Medicine, Nagasaki University, Nagasaki 852-8523, Japan; 5Medical Education Development Center, Nagasaki University Hospital, Nagasaki 852-8501, Japan; 6Infection Control and Education Center, Nagasaki University Hospital, Nagasaki 852-8501, Japan; 7Department of Infectious Diseases, Nagasaki University Graduate School of Biomedical Sciences, Nagasaki 852-8501, Japan; 8Clinical Oncology Center, Nagasaki University Hospital, Nagasaki 852-8501, Japan; 9Department of General Medicine, Nagasaki University Hospital, Nagasaki 852-8501, Japan; 10Department of Infectious Diseases, Nagasaki University Hospital, Nagasaki 852-8523, Japan; 11Infectious Diseases Experts Training Center, Nagasaki University Hospital, Nagasaki 852-8501, Japan; 12Department of Laboratory Medicine, Nagasaki University Hospital, Nagasaki 852-8501, Japan

**Keywords:** COVID-19, outpatient, triage checklist, emergency department visit, hospitalization

## Abstract

Managing mild illness in COVID-19 and predicting progression to severe disease are concerning issues. Here, we investigated the outcomes of Japanese patients with mild COVID-19, and identified triage risk factors for further hospitalization and emergency department (ED) visits at a single tertiary hospital. A triage checklist with 30 factors was used. Patients recommended for isolation were followed up for 10 days for subsequent ED visits or hospital admission. Overall, 338 patients (median age, 44.0; 45% women) visited the clinic 5.0 days (median) after symptom onset. Thirty-six patients were immediately hospitalized following triage; others were isolated. In total, 72 non-hospitalized patients visited the ED during their isolation, and 30 were hospitalized after evaluation for oxygen desaturation. The median ED visit and hospitalization durations after symptom onset were 5.0 and 8.0 days, respectively. The checklist factors associated with hospitalization during isolation were age > 50 years, body mass index > 25 kg/m^2^, hypertension, tachycardia with pulse rate > 100/min or blood pressure > 135 mmHg at triage, and >3-day delay in hospital visit after symptom onset. No patients died. Altogether, 80% of patients with mild COVID-19 could be safely isolated at home. Age, BMI, underlying hypertension, date after symptom onset, tachycardia, and systolic blood pressure at triage might be related to later hospitalization.

## 1. Introduction

In January 2020, coronavirus disease 2019 (COVID-19)—caused by the severe acute respiratory syndrome coronavirus 2 (SARS-CoV-2)—first appeared in Japan, and it has caused five pandemic waves as of the end of 2021. Overall, 2.93 million polymerase chain reaction (PCR)-positive cases have been confirmed, and 18,393 deaths have been reported by the Ministry of Health, Labor, and Welfare in Japan as of 31 December 2021 [1]. During the COVID-19 pandemic, emergency departments (EDs) were involved in the crisis, managing cases of severe respiratory failure as well as large numbers of patients with mild symptoms [2]. This caused overcrowding in many EDs and the unavailability of hospital beds across the nation [3]. An early triage system is critical for identifying patients at a high risk of severe COVID-19 who will require hospitalization [4]. To date, many risk factors have been identified that contribute to the severity of COVID-19, such as advanced age, obesity, and diabetes mellitus [5]. However, there is minimal evidence regarding the outcomes of patients with mild COVID-19 who are not hospitalized and are under home or facility isolation. Factors associated with later exacerbation of symptoms, leading to an ED visit and admission of these patients, have not been clarified. Although more than 80% of patients with COVID-19 have mild disease [6], this percentage has further increased following the introduction of SARS-CoV-2 vaccinations [7]. Investigation of patients with mild disease is important to avoid the disruption of EDs and society at large. The present study aimed to assess the outcomes of patients with mild COVID-19 in terms of subsequent ED visits and hospitalization following the initial triage, and to identify the factors leading to these outcomes.

## 2. Materials and Methods

### 2.1. Study Population

This was a retrospective, observational cohort study of patients aged ≥20 years, defined as adults in Japan, who visited the COVID-19 triage outpatient clinic at Nagasaki University Hospital (NUH) between 1 April 2021 and 31 May 2021 during Japan’s “fourth wave” of the COVID-19 pandemic, mainly caused by the clade 20I/1.1.7/A Alpha. Since the pandemic, a COVID-19 triage outpatient clinic has been operational at NUH for patients with mild illness. SARS-CoV-2 PCR-positive patients detected in the city of Nagasaki are listed by the Nagasaki City Public Health Center and reported to the Nagasaki University Hospital Infection Control and Education Center, which then assigns these patients to hospitals for initial assessment, or sends them directly to EDs. NUH caters to the largest number of patients with COVID-19 in the city of Nagasaki. Such patients arriving at NUH for their first assessment are examined by a physician in a triage room or tent and interviewed using a COVID-19 checklist while a nurse records their vital signs. Patients who had already started oxygen therapy at the initial institution, along with those who were initially triaged with the necessity of hospitalization, directly visited, or transported to the emergency room for hospital admission without outpatient triage, were excluded from this analysis. All isolated patients with COVID-19 in the city of Nagasaki were followed up for health observations by phone for at least 10 days by the Nagasaki City Public Health Center, which coordinated medical visits if necessary.

### 2.2. Factors in the Triage Checklists

Checklist factors included age, sex, date of symptom onset, height, weight, body mass index (BMI), vital signs (i.e., body temperature (BT), respiratory rate (RR), SpO_2_, pulse rate (PR), systolic blood pressure (SBP), diastolic blood pressure (DBP)), underlying diseases (i.e., hypertension, diabetes, cardiovascular diseases, chronic lung diseases, chronic liver diseases, chronic renal diseases, malignancies, history of organ transplantation, human immunodeficiency virus (HIV) infection, use of oral steroids/immunosuppressive drugs, and pregnancy), and subjective symptoms that could be considered for hospitalization (e.g., fatigue with difficulty moving, dyspnea, chest pain, poor dietary intake, and severe diarrhea with ≥6 bowel movements per day). The checklist was created by including factors for evaluation of general condition to determine the indication for hospitalization, in addition to the risk factors for severe COVID-19 that are described in the Japanese COVID-19 practice guidelines [8]. Indication for hospitalization was determined by a physician based on the checklist. Hospitalization was recommended if the patient’s SpO_2_ was ≤95%, RR was ≥22/min, SBP was ≤90 mmHg, or if severe dyspnea was present.

### 2.3. Genetic Background of SARS-CoV-2 in Nagasaki Prefecture during the Study Period

The epidemic since March 2021 in Japan has been referred to as the “fourth wave”. To clarify the genotypic changes in the fourth wave, we constructed a phylogenetic tree of SARS-CoV-2 isolated in Nagasaki Prefecture. Overall, 1810 records of SARS-CoV-2 genome sequences from Nagasaki Prefecture were available on GISAID [9] as of 24 March 2022. The tree was generated by Nextstrain [10,11]. The ‘third wave’ in Japan was started from October 2020 until the end of February 2021. COVID-19 was not endemic in Nagasaki Prefecture after the “third wave” until the end of March 2021. The phylogenetic tree (Appendix A) suggested that the clade 20B/R.1 had remained in Nagasaki Prefecture during the low endemic period. The “fourth wave” was mainly caused by the clade 20I/B.1.1.7/Alpha. This clade was first detected in Japan in December 2020, and was introduced into Nagasaki Prefecture until late March 2021. The tree suggests that at least four independent subclades of 20I/B.1.1.7/Alpha were introduced into Nagasaki Prefecture around the end of March 2021. In addition, the old clade, 20B/R.1, remained in Nagasaki Prefecture until May 2021. The city of Nagasaki is in the southern part of Nagasaki Prefecture. The diversity of genomic RNA sequences suggested that 20I/B.1.1.7/Alpha spread rapidly in Nagasaki Prefecture, beginning in early April 2021. However, many of the GISAID records do not include geographical information in Nagasaki Prefecture. Therefore, it is possible that some of the cases investigated in this study may have been caused by the older 20B/R.1.

### 2.4. Endpoint

The outcomes of the present study were (1) ED visit during isolation and (2) hospitalization following an ED visit, during a minimum follow-up duration of 10 days.

### 2.5. Ethics

This study was approved by the Ethical Review Committee of NUH (permission number 21101927), and was conducted in accordance with the Declaration of Helsinki. The research was conducted in an opt-out format, and the information was disclosed on the website of the Clinical Research Center, NUH (http://www.mh.nagasaki-u.ac.jp/research/rinsho/patients/intmed-2/202111_01.pdf, accessed on 16 December 2021).

### 2.6. Analysis

All variables in the COVID-19 triage checklist were included in the analysis. Events were designated as “ED visit during isolation” or “hospitalization following ED visit”, and the association of a variable with each of these events was described as the odds ratio (OR) or the median difference. Days post-onset of symptoms, age, BMI, BT, RR, SpO_2_, PR, and SBP were converted to categorical values by using cutoff values (≥3 days, ≥25.0 kg/m^2^, ≥38.0 °C, ≥22/min, <94%, ≥100/min, and ≥135 mmHg, respectively). The *p*-values were used to screen variables.

## 3. Results

### 3.1. Patients’ Characteristics at Outpatient Triage

#### 3.1.1. Baseline Characteristics

The period of the study covered the “fourth wave” of the COVID-19 pandemic in Japan, which was caused mostly by the clade 20I/B.1.1.7/Alpha, and partially by the old clade 20B/R.1 (Appendix A). Overall, 338 SARS-CoV-2-positive patients visited the NUH outpatient triage clinic, and their characteristics are listed in Table 1. This study group included six healthcare workers. All patients were of Japanese ethnicity, with a median age of 44.0 years, and 45% of them were women. Outpatient triage visits occurred at a median of 5.0 days after symptom onset. The median values for BMI, BT, RR, SpO_2_, PR, and SBP/DBP at triage were 22.9 kg/m^2^, 36.9 °C, 16.0/min, 97.0%, 86.0/min, and 128/77 mmHg, respectively. Overall, 71 patients (21.0%) had underlying diseases. The most frequent comorbidity was hypertension (11%), followed by chronic lung diseases (4.4%), cardiovascular diseases (3.8%), and diabetes (3.6%). Of all patients, 2.1% were pregnant. None of the patients had a history of organ transplant or HIV infection. Symptoms that potentially necessitated hospitalization were found in 111 patients (32.8%). Poor dietary intake (22.0%) was the most frequent symptom calling for hospitalization, followed by dyspnea (8.3%) and chest pain (6.5%). Of the 338 patients evaluated, 36 patients (10.6%) were hospitalized at the initial triage, while the remaining 302 patients were isolated in facilities or at home (Figure 1).

#### 3.1.2. Factors Associated with Immediate Hospitalization

To assess the factors in the checklist relating to admission of patients, the group of patients hospitalized at initial triage was compared with those who underwent facility or home isolation. The factors that were associated (screened using *p*-values at *p* < 0.1) with immediate hospitalization at triage are presented in Table 2. The median time of triage visit after symptom onset was 5.0 days. The former group of patients was also considerably older (median, 58.5 years) and had a higher BMI (median, 26.5 kg/m^2^) than the non-hospitalized group. Five patients were taking immunosuppressants—including steroids, tacrolimus, methotrexate, or etanercept—for connective tissue diseases, nephritis, or bronchial asthma. Moreover, the following clinical factors were associated with hospitalization at triage: high fever, tachypnea, oxygen desaturation, and tachycardia. BT was the only factor that did not overlap between the hospitalized and non-hospitalized groups. Comorbidities, such as hypertension, diabetes, cardiovascular diseases, or liver diseases, were associated with hospitalization at triage. Patients hospitalized at triage had significant symptoms of severe fatigue, dyspnea, or poor dietary intake. The outcomes of patients who were immediately hospitalized at initial triage were as follows: 3 patients (8.3%) did not progress from mild disease; 11 patients (30.6%) progressed to non-oxygen-requiring pneumonia; 21 patients (58.3%) progressed to moderate pneumonia requiring oxygen supplementation; and 1 patient (2.8%) died from severe disease (Figure 1).

### 3.2. Triage Factors Associated with ED Visit during Isolation in Facilities or at Home

Among the 302 patients who were isolated in facilities or at home, 71 patients (23.5%) visited the ED during the isolation period (Figure 1). The median duration from symptom onset to ED visit was 5.0 days. Patients who visited the ED while in isolation and those who did not visit the ED while in isolation were compared. Table 3 lists the factors that were likely associated (screened using *p*-values at *p* < 0.1) with ED visits while being in facility or home isolation. ED visits during isolation were associated with older age (median, 50.0 vs. 40.0 years). A total of 16.9% of patients had underlying hypertension, while 12.7% reported having chest pain at triage, and these were more frequently found in patients who visited the ED during isolation.

Prescriptions and their association with ED visits were also examined. Overall, 218 patients were prescribed acetaminophen, 77 patients were prescribed cough suppressants, 10 patients were prescribed NSAIDs, and 4 patients were prescribed antiviral agents as part of a clinical trial. No patients were prescribed anticoagulants or platelet anti-aggregates. Of these patients, ED visits were reported for 50 patients who were prescribed acetaminophen (OR, 0.89 [95% CI, 0.50–1.60]), 21 patients who were prescribed cough suppressants (OR, 1.31 [95% CI, 0.73–2.37]), 1 patient who was prescribed NSAIDs (OR, 0.35 [95% CI, 0.04–2.83]), and 1 patient who was prescribed antiviral agents (OR, 1.09 [95% CI, 0.11–10.60]).

### 3.3. Triage Factors Associated with Hospital Admission at ED Visit during Isolation in Facilities or at Home

Among the 302 patients who were isolated in facilities or at home after triage, 30 patients (9.9%) were hospitalized after an ED visit during the isolation period (Figure 1). The main reason for hospitalization was oxygen desaturation, which occurred in 21 of those 30 patients (70%). To assess which checklist factors at triage distinguished patients who required hospitalization during isolation, we compared the characteristics of patients who were hospitalized at ED visits and those who were not hospitalized at ED visits. Checklist factors that were likely associated (screened by the *p*-values at *p* < 0.1) with hospitalization at ED visits during the isolation period are presented in Table 4. The median duration from COVID-19 symptom onset to hospitalization was 8.0 days. Patients hospitalized at ED visits were likely to present at initial triage later (5.0 days) than those in the non-hospitalized post-symptomatic group (4.0 days). The median duration from the first triage to the ED visit was 5.0 days. Patients who were hospitalized following ED visits during isolation were older (median, 58.5 vs. 39.0 years) and had a higher BMI (median, 25.3 vs. 23.5 kg/m^2^). At triage, patients who were hospitalized following ED visits during isolation had low-grade fever (median, 37.5 vs. 37.0 °C) and higher SBP (median, 140.0 vs. 130.0 mmHg). A total of 30% of patients who were hospitalized at ED visits had underlying hypertension. The outcomes of patients hospitalized following ED visits were as follows: none maintained their mild disease (0%); 9 (30.0%) progressed to non-oxygen-requiring pneumonia; 21 (70.0%) progressed to oxygen-requiring pneumonia; and none died or progressed to a severe disease requiring intensive care unit admission (Figure 1).

A heatmap for variables comparing those who were hospitalized at ED visits (*n* = 30) to those who were not hospitalized (*n* = 272) during isolation is shown in Figure 2. The odds ratio (OR) of hospitalization was obtained for each variable. The continuous variables were dichotomized at the cutoffs stated by the Youden index. Age > 50 years (OR = 6.8), obesity with BMI > 25 kg/m^2^ (OR = 3.2), underlying hypertension (OR = 6.4), tachycardia with PR  > 100/min (OR = 2.7), high BP (SBP  > 135 mmHg) at triage (OR = 3.5), and delay of presence at hospital of >3 days after symptom onset (OR = 3.5) were associated with hospitalization during the isolation period. To consider confounding and other variables related to delay of triage visits (“duration” in Figure 2), the distribution of the delay was explored using the relative rank of the number of days. The results showed that patients aged ≥65 years and of male sex had a tendency for later triage visits (median, 1.5 days) compared to those aged <65 years or of the female sex (median, 1.0 days) (Appendix A).

We developed a scoring system consisting of a linear combination of predictors and their weights for readmission. The predictors and weights were determined using the Akaike information criterion (AIC) of the model, which was minimized by the forward–backward stepwise method. The candidate predictors were the variables shown in Figure 2. As a result, the following predictors were included in the final model: age, gender, BMI, pulse rate, hypertension, diabetes, and lung disease. The weights were 1.7832 for age, 0.8377 for gender (female = 0, male = 1), 0.7590 for BMI, 8.8727 for pulse rate, 1.3611 for having hypertension, −1.8869 for having diabetes, and −15.3417 for having lung disease; the intercept was −4.1885. The accuracy in the prediction using the same data used in the model development is shown in Appendix A. The sensitivity and specificity in the prediction were calculated by moving the threshold for the score (*x*-axis).

This prediction model was applied to another COVID-19 outpatient-triaged population consisting of a total of 267 patients in the “third wave” (December 2020 through January 2021). The background and characteristics of this population were as follows (all median values): age, 44 years; female, 53.2%; BMI, 22.8 kg/m^2^; BT, 36.7 °C; RR, 15.0/min; SpO_2_, 98%; HR, 83/min; SBP, 128 mmHg; and duration after onset, 5 days. There were 27 patients with hypertension, 11 with diabetes, 5 with immunosuppressed conditions, 3 with malignancies, 9 with cardiovascular diseases, 12 with lung diseases, and 4 with liver diseases. In total, 18 patients were hospitalized at the initial triage, 43 patients visited the ED during the isolation period, and 18 patients were hospitalized following the ED visit. The accuracy in the predictions by the predictive model developed is shown by external validation plots in Appendix A, showing that the sensitivities decreased for predicting the admission using the same threshold of the score. The “third wave” was caused mainly by the 20B/R.1 variant (Appendix A), whose clinical characteristics may be different from those of the 20I/B.1.1.7/Alpha variant in the “fourth wave”; therefore, it would be difficult to apply the predictive equation between populations infected with different variants.

## 4. Discussion

In this retrospective cohort study of 338 SARS-CoV-2-positive patients, we found evidence to suggest that relatively young patients (age < 50 years) without obesity, triaged within 3 days after symptom onset, and with no tachycardia or no hypertension at triage, can self-manage their conditions without hospitalization. Our sample had a higher rate of ED visits than the 14% rate reported by a previous study, which followed up patients with coronavirus-like symptoms during the COVID-19 pandemic [12]. Our study results show that >10% of the patients were hospitalized at the initial triage visit, and that another 10% were hospitalized following an ED visit during isolation; these proportions were similar when compared to previous reports [13,14]. One patient died in the immediate hospitalization group, and none of the patients died after hospitalization during isolation. It was found that the mortality rate of all patients with COVID-19 in the entire cohort was 2.95%—a high rate compared to public health data for the alpha variant (0.3–6%), although variable between reports [15]. The mortality rate of isolated patients with COVID-19 after triage was 0%. Concerning the hospitalization rate during isolation, the 10% observed in our cohort was high compared to recent studies in Japan (3.5–6.4%) [16,17]. The high prevalence of disease progression and hospitalization in our cohort may have been caused by the alpha variant [18] and the higher age of our cohort population. Factors associated with hospitalization during isolation in our study were consistent with other reports, which again highlighted older age and higher BMI [16,17]. The clinical course of disease progression to ED visit or hospitalization in our cohort was similar to that reported in previous publications, which described progression to maximal symptom severity at 8–10 days after symptom onset [19]. The Ministry of Health, Labor, and Welfare of Japan has specified that an isolation period of at least 10 days is required after the onset of COVID-19 symptoms to prevent transmission [20], which is a reasonable time for patients to observe their symptoms for possible disease progression. In our study, the mean interval from first triage to ED return was 2 days, which is much shorter than the period of 5 days reported previously [12]. We attribute this result to the dense monitoring system of Nagasaki City Public Health Center and its isolation facilities. Our results suggested that a later triage visit after symptom onset was associated with ED visits or hospitalization during the isolation period. Similar to previous reports [21,22], a delay in hospital visits was observed in patients who were aged >65 years or were male. This finding may reflect the difference in health-seeking behavior across sex and age groups, and/or the non-recognition of developing symptoms in the elderly [23]. Our study indicated that patients with COVID-19 aged 50 years or over, those with obesity (BMI > 25 kg/m^2^), those with underlying hypertension, those presenting to the hospital >3 days after symptom onset, and those presenting with tachycardia (PR > 100/min) [2] or high BP (SBP > 135 mmHg) at triage significantly required admission during isolation. Outpatients with COVID-19 with the abovementioned features at first triage should be followed up remotely [24] or by phone [25] to prevent them from missing their proper hospital visit.

This study has several limitations. First, this was a retrospective study and, therefore, has a potential for biases from incomplete clinician documentation. Second, follow-up information was not available after patients were discharged, which may have led to underestimated rates of hospitalization or mortality. Third, members of the team reviewing the ED revisits were not blinded, which may have introduced assessment bias. Fourth, since different variants of coronavirus have different clinical characteristics, important factors at triage may be different in other pandemic waves caused by different variants; thus, it may not be possible to simply apply the predictive score to other patient populations. Fifth, the number of patients evaluated in this study was small, and may lack statistical power. Finally, the generalizability of the present study’s findings to other settings is limited by its single-center design. Because of site-specific characteristics, it is possible that other ED sites with differences in the availability of transportation and access to professional medical support may have different rates of outcomes than our own patient cohort. Even with these limitations, this study may contribute to the literature by evaluating for the first time the potential benefits of a triage checklist in avoiding treatment delays and subsequent hospitalization of patients with mild COVID-19.

## 5. Conclusions

In summary, approximately 80% of patients with mild COVID-19 can be safely isolated at home or in a facility. Approximately 10% of patients will experience progression of symptoms in the ensuing week that will require hospitalization for treatment—typically at 10 days of symptomatic illness. Clinicians should inform patients—especially those aged >50 years, with BMI > 25 kg/m^2^, with underlying hypertension, or presenting to the hospital >3 days after symptom onset with tachycardia (PR > 100/min) or high BP (SBP > 135 mmHg) at triage—that they may experience worsening of symptoms after their first visit and eventually require hospitalization. Such patients should be advised to seek follow-up assessment by a medical professional. Further prospective studies in the upcoming COVID-19 pandemic waves are recommended to validate checklist factors and cutoff values that affect patients’ outcomes.

## Figures and Tables

**Figure 1 jcm-11-05444-f001:**
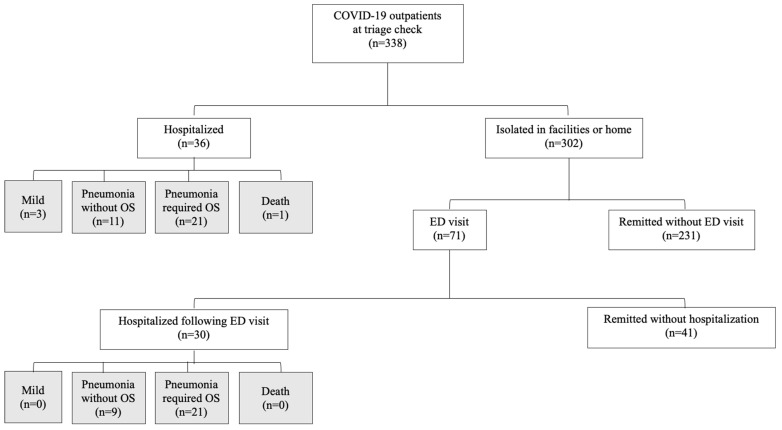
Flowchart demonstrating the numbers of patients with COVID-19 who visited the outpatient triage, who were immediately hospitalized at triage, who visited the ED during isolation, and who were hospitalized upon their ED visit during isolation (white-filled squares). Hospitalized patients were followed up, and their outcomes were recorded (gray-filled squares). Abbreviations: OS, oxygen supplementation; ED, emergency department.

**Figure 2 jcm-11-05444-f002:**
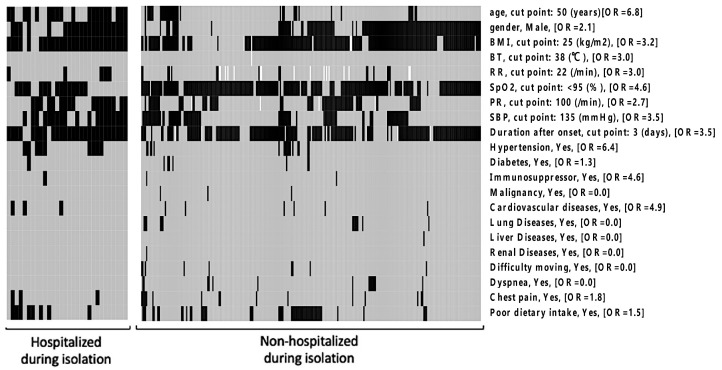
Heatmap of each variable in the COVID-19 outpatient triage: Patients hospitalized at ED visit (*n* = 30) and those without hospitalization during isolation (*n* = 272) were grouped. Black-filled patients represent values over cutoff values (for continuous variables) or the presence of underlying diseases or symptoms. Gray-filled patients represent values lower than the cutoff values (for continuous variables) or the absence of underlying diseases or symptoms. White-filled patients represent missing data. The cutoff values were defined using the Youden index. In the brackets, ORs for the association between the variable (≤median, >median) and the occurrence of “Hospitalized at ED visit” are mentioned. Abbreviations: BMI, body mass index; BT, body temperature; ED, emergency department; OR, odds ratio; PR, pulse rate; RR, respiratory rate; SBP, systolic blood pressure.

**Table 1 jcm-11-05444-t001:** Characteristics of patients with COVID-19 and the checklist factors at outpatient triage.

Checklist Factors	Patients with COVID-19 at Triage (*n* = 338)	Checklist Factors	Patients with COVID-19 at Triage (*n* = 338)
Days post-onset of symptoms	5.0 [3.3; 7.0]	RR (/min)	16.0 [16.0; 20.0]
Age	44.0 [32.0; 56.0]	SpO_2_ (%)	97.0 [96.0; 98.0]
Female sex	152 (45.0)	PR (/min)	86.0 [78.0; 98.0]
BMI (kg/m^2^)	22.9 [20.8; 25.8]	SBP (mmHg)	128.0 [114.0; 142.0]
BT (°C)	36.9 [36.5; 37.4]	DBP (mmHg)	77.0 [67.0; 85.0]
Underlying diseases	71 (21.0)	Chronic renal diseases	2 (0.6)
Hypertension	37 (11.0)	Malignancy	5 (1.4)
Diabetes mellitus	12 (3.6)	Organ transplant	0 (0)
Cardiovascular diseases	13 (3.8)	HIV infection	0 (0)
Chronic lung diseases	15 (4.4)	Other immunosuppressed status	5 (1.4)
Chronic liver diseases	3 (0.9)	Pregnancy	7 (2.1)
Symptoms with consideration for hospitalization	111 (32.8)	Chest pain	22 (6.5)
Fatigue with difficulty moving	14 (4.1)	Poor dietary intake	76 (22.0)
Dyspnea	28 (8.3)	Severe diarrhea (>6 times/day)	6 (1.8)

Abbreviations: BMI, body mass index; BT, body temperature; DBP, diastolic blood pressure; ED, emergency department; HIV, human immunodeficiency virus; PR, pulse rate; RR, respiratory rate; SBP, systolic blood pressure; Q1, first quartile; Q3, third quartile. Data are shown as medians [Q1; Q3] or *n* (%).

**Table 2 jcm-11-05444-t002:** Checklist factors and immediate hospitalization at outpatient triage.

Checklist Factors	Immediate Hospitalization (*n* = 36)	Isolation in Facilities or Home (*n* = 302)	Odds Ratio (95% CI)
Days post-onset of symptoms	5.0 [4.0; 8.0]	5.0 [3.0; 7.0]	
≥3 days	33 (91.7)	272 (90.1)	1.52 (0.42–9.67)
Age	58.5 [50.3; 67.0]	41.0 [31.0; 53.3]	
≥50 years	29 (80.6)	100 (33.1)	8.37 (3.74–21.37)
BMI (kg/m^2^)	26.5 [22.7; 28.7]	22.7 [20.5; 25.4]	
≥25.0 kg/m^2^	20 (55.6)	85 (28.1)	3.31 (1.63–6.87)
BT (°C)	37.6 [36.9; 38.6]	36.9 [36.5; 37.3]	
≥38.0 °C	15 (41.7)	23 (7.6)	8.63 (3.90–19.03)
RR (/min)	18.0 [15.8; 26.5]	16.0 [15.5; 20.0]	
≥22/min	14 (38.9)	28 (9.3)	6.45 (2.91–14.16)
SpO_2_ (%)	95.0 [94.0; 97.0]	97.0 [97.0; 98.0]	
<94%	13 (36.1)	3 (1.0)	55.96 (16.67–257.20)
PR (/min)	94.0 [84.3; 105.8]	85.0 [78.0; 96.0]	
≥100/min	14 (38.9)	57	2.69 (1.27–5.54)
Hypertension	11 (30.6)	26 (8.6)	4.67 (2.01–10.42)
Cardiovascular diseases	4 (11.1)	9 (3.0)	4.07 (1.06–13.28)
Diabetes mellitus	4 (11.1)	8 (2.6)	4.59 (1.17–15.46)
Chronic liver diseases	2 (5.6)	1 (0.3)	17.70 (1.65–386.93)
Severe fatigue with hard to move	5 (13.9)	9 (3.0)	5.25 (1.53–16.22)
Malignancy	2 (5.6)	3 (1.0)	5.86 (0.75–36.57)
Immunosuppressed status	2 (5.6)	3 (1.0)	5.86 (0.75–36.57)
Dyspnea	14 (38.9)	14 (4.6)	13.09 (5.55–31.28)
Poor dietary intake	14 (38.9)	62 (20.5)	2.46 (1.17–5.05)

Abbreviations: BMI, body mass index; BT, body temperature; CI, confidence interval; PR, pulse rate; RR, respiratory rate; Q1, first quartile; Q3, third quartile. Data are shown as medians [Q1; Q3] or *n* (%).

**Table 3 jcm-11-05444-t003:** Checklist factors and ED visits during isolation in facilities or at home.

Checklist Factors	ED Visit (*n* = 71)	Remission without ED Visit (*n* = 231)	Odds Ratio (95% CI)
Age	50.0 [35.0; 64.0]	40.0 [29.0; 50.0]	
≥50 years	37 (52.1)	63 (27.3)	2.90 (1.68–5.04)
BMI (kg/m^2^)	24.4 [21.5; 26.1]	22.3 [20.2; 24.9]	
≥25.0 kg/m^2^	29 (40.8)	56 (24.2)	2.15 (1.22–3.77)
BT (°C)	37.1 [36.7; 37.7]	36.8 [36.4; 37.2]	
≥38.0 °C	10 (17.1)	13 (5.6)	2.79 (1.14–6.67)
SpO_2_ (%)	97.0 [96.0; 98.0]	97.0 [97.0; 98.0]	
<94%	1 (1.4)	2 (0.9)	1.65 (0.08–17.50)
PR (/min)	87.0 [79.0; 100.0]	84.0 [77.0; 95.0]	
≥100/min	18 (25.4)	39 (16.9)	1.64 (0.85–3.06)
SBP (mmHg)	134.0 [121.0; 147.0]	126.0 [113.0; 139.0]	
≥135 mmHg	35 (49.3)	68 (29.4)	2.32 (1.34–4.00)
Underlying Hypertension	12 (16.9)	14 (6.0)	3.15 (1.37–7.19)
Chest pain	9 (12.7)	10 (4.3)	3.21 (1.22–8.30)

Abbreviations: BMI, body mass index; BT, body temperature; CI, confidence interval; ED, emergency department; PR, pulse rate; SBP, systolic blood pressure; Q1, first quartile; Q3, third quartile. Data are shown as medians [Q1; Q3] or *n* (%).

**Table 4 jcm-11-05444-t004:** Checklist factors associated with hospitalization following ED visits during isolation.

	Hospitalized at ED (*n* = 30)	Non-Hospitalized at ED (*n* = 41)	Odds Ratio (95% CI)
Days post-onset of symptoms	5.0 [4.0; 6.3]	4.0 [3.0; 6.0]	NA (* -NA)
≥3 days	30 (100.0)	35 (85.4)	
Age	58.5 [48.8; 70.3]	39.0 [32.0; 530]	
≥50 years	22 (73.3)	15 (36.6)	4.77 (1.76–13.97)
BMI (kg/m^2^)	25.3 [21.9; 30.0]	23.5 [21.1; 25.9]	
≥25.0 kg/m^2^	16 (53.3)	13 (31.7)	2.37 (0.90–6.43)
BT (°C)	37.5 [37.0; 37.9]	37.0 [36.5; 37.4]	
≥38.0 °C	7 (23.3)	3 (7.3)	3.75 (0.94–18.79)
SpO_2_ (%)	96.5 [96.0; 97.0]	97.0 [97.0; 98.0]	
<94%	1 (3.3)	0 (0.0)	NA (* -NA)
PR (/min)	94.5 [81.8; 104.3]	84.0 [78.0; 97.5]	
≥100/min	10 (33.3)	8 (19.5)	2.06 (0.70–6.26)
SBP (mmHg)	140.0 [124.0; 149.8]	130.0 [112.5; 140.5]	
≥135 mmHg	19 (63.3)	16 (39.0)	2.70 (1.04–7.32)
Underlying hypertension	9 (30.0)	3 (7.3)	5.43 (1.44–26.52)

Abbreviations: ED, emergency department; BMI, body mass index; BT, body temperature; PR, pulse rate; SBP, systolic blood pressure; Q1, first quartile; Q3, third quartile. Data are shown as medians [Q1; Q3] or *n* (%). Odds ratios and 95% CIs are shown as OD (95% CI); *p*-Values < 0.01 are indicated by *.

## Data Availability

Data are contained within the article.

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
