# Peer review of "Evaluation of a Triage Checklist for Mild COVID-19 Outpatients in Predicting Subsequent Emergency Department Visits and Hospitalization during the Isolation Period: A Single-Center Retrospective Study"

_jcm, 2022, doi:10.3390/jcm11185444_

Round 1

Reviewer 1 Report

This was an interesting study investigating the impact of a triage checklist on the management of Covid-19. Although there are many publications about the management of Covid-19 at hospitals, outpatient management strategies are not described in detail. The study underlines that if patients with underlying diseases or clinical conditions which increase the risk for severe Covid-19 admit earlier with mild disease, the attending physician should be alerted to the increased risk of severe Covid-19 and should inform the patient. This was the main approach for many Covid-19 outpatient clinics and was well-documented by the authors of this single-center study. 

There are some points that can improve the paper:

1. Developing a score based on the risk factors  for re-admission can be interesting

2.  Were the patients who were admitted to the outpatient clinic prescribed any antiviral agents,  NSAIDs, anticoagulants, or antiaggregants? If yes, what was the influence of these treatments on re-admission?

3. Were three any healthcare workers in the study groups? 

Author Response

Reviewer: 1

This was an interesting study investigating the impact of a triage checklist on the management of Covid-19. Although there are many publications about the management of Covid-19 at hospitals, outpatient management strategies are not described in detail. The study underlines that if patients with underlying diseases or clinical conditions which increase the risk for severe Covid-19 admit earlier with mild disease, the attending physician should be alerted to the increased risk of severe Covid-19 and should inform the patient. This was the main approach for many Covid-19 outpatient clinics and was well-documented by the authors of this single-center study. 

There are some points that can improve the paper:

Response: Thank you for reviewing our manuscript and for providing insightful comments. We appreciate your positive feedback. We have revised our manuscript to the best of our ability according to your suggestions. Please find our point-by-point responses below.

  1. Developing a score based on the risk factors for re-admission can be interesting

Response: Thank you for your valuable comments. Per your comment, we have created a scoring system based on the results of the study and applied it to another population, as follows (lines 271–296):

“We developed a scoring system consisting of a linear combination of predictors and their weights for re-admission. The predictors and weights were determined as the Akaike’s Information Criterion (AIC) of the model, which was minimized by the forward-backward stepwise method. The candidate predictors were the variables shown in Figure 2. As a result, the following predictors were included in the final model: age, gender, BMI, pulse rate, hypertension, diabetes, and lung disease. The weights were 1.7832 for age, 0.8377 for gender (female=0, male=1), 0.7590 for BMI, 8.8727 for pulse rate, 1.3611 for having hypertension, -1.8869 for having diabetes, and -15.3417 for having lung disease; the intercept = -4.1885. The accuracy in the prediction using the same data used in the model development is shown in Figure S3 and Figure S4. The sensitivity and specificity in the prediction were calculated by moving the threshold for the score (x axis).

This prediction model was applied to another COVID-19 outpatient triaged population consisting of a total of 267 patients in the “third wave” (December 2020 through January 2021). The background and characteristics of this population were as follows (all median values): age, 44 years; female, 53.2%; BMI, 22.8 kg/m2; BT, 36.7°C; RR, 15.0/min; SpO2, 98%; HR, 83/min; SBP, 128 mmHg; and duration after onset, 5 days. There were 27 patients with hypertension, 11 with diabetes, 5 with immunosuppressed conditions, 3 with malignancy, 9 with cardiovascular disease, 12 with lung diseases, and 4 with liver diseases. A total of 18 patients were hospitalized at the initial triage, 43 patients visited the ED during the isolation period, and 18 patients were hospitalized following the ED visit. The accuracy in the prediction by the predictive model developed are shown as external validation plot in Figure S3 and Figure S4. It shows that the sensitivities decreased for predicting the admission using the same threshold of the score. The “third wave” was caused mainly by the 20B/R.1 variant (Figure S1), whose clinical characteristics may be different from those of the 20I/B.1.1.7/Alpha variant in the “fourth wave”; therefore, it would be difficult to apply the predictive equation between populations infected with different variants.”

However, since the clinical profile in COVID-19 differs based on factors such as vaccination status or SARS-CoV-2 variants, the score obtained in this study cannot simply be applied to other patients with different SARS-CoV-2 variant infections. This limitation has been added to lines 352–356 as follows:

“Fourth, since different variants of coronavirus have different clinical characteristics, important factors at the triage may be different in other pandemic waves caused by different variants, and thus, it may not be possible to simply apply the predictive score to other patient populations.”

  1. Were the patients who were admitted to the outpatient clinic prescribed any antiviral agents, NSAIDs, anticoagulants, or anti-aggregants? If yes, what was the influence of these treatments on re-admission?

Response: Thank you for your valuable comment. Most prescriptions were for acetaminophen and cough suppressants, NSAIDs, and anti-viral agents. No anticoagulants or platelet anti-aggregates were prescribed in the outpatient clinic during follow-ups. Information on prescriptions for patients has been added to lines 219–227 as follows:

“Prescriptions and association with ED visit were also examined. Overall, 218 patients were prescribed acetaminophen, 77 patients were prescribed cough suppressants, 10 patients were prescribed NSAIDs, and 4 patients were prescribed anti-viral agents as part of a clinical trial. No patients were prescribed anti-coagulants or platelet anti-aggregates. Of these patients, ED visits were reported for 50 patients who were prescribed acetaminophen (OR, 0.89 [95% CI, 0.50–1.60]), 21 patients who were prescribed cough suppressants (OR, 1.31 [95% CI, 0.73–2.37]), 1 patient who was prescribed NSAIDs (OR, 0.35 [95% CI, 0.04–2.83]), and 1 patient who was prescribed anti-viral agents (OR, 1.09 [95% CI, 0.11–10.60]).”

  1. Were three any healthcare workers in the study groups? 

Response: Thank you for your comment. There were six healthcare workers in our study group. This information has been added to lines 160–161 as follows: “This study group included six healthcare workers.”

Reviewer 2 Report

The manuscript is well written and the topic is emerging. There are some minor comments to improve the abstract. 

Abstract

1) Abstract can not be started directly with the aim of study, At least one sentence of introductory background is needed. 

2) It would be helpful if authors revise the abstract in a structured format of introduction, method, result and conclusion

3) Conclusion is too short not effectively written, it needs to be improved in consistent with the conclusion of manuscript. 

Author Response

Reviewer: 2

The manuscript is well written and the topic is emerging. There are some minor comments to improve the abstract. 

Response: Thank you for reviewing our manuscript. We appreciate your positive feedback. Please find our point-by-point responses below.

Abstract

1) Abstract cannot be started directly with the aim of study, At least one sentence of introductory background is needed. 

Response: Thank you for your comment. We have added an introductory sentence to the abstract as follows (lines 43–44): “Managing mild illness of COVID-19 and predicting progression to severe disease are concerning issues.”

2) It would be helpful if authors revise the abstract in a structured format of introduction, method, result and conclusion

Response: Thank you for your valuable comment. We have revised the abstract per journal instructions (https://www.mdpi.com/journal/jcm/instructions), which state that “The abstract should be a single paragraph and should follow the style of structured abstracts, but without headings.”

3) Conclusion is too short and not effectively written, it needs to be improved in consistent with the conclusion of manuscript. 

Response: Thank you for your valuable comment. Owing to the word count limitation of 200 words for the abstract per journal instructions, we are unable to add more details. We have revised the conclusion in the abstract as follows (lines 56–58): “Altogether, 80% of patients with mild COVID-19 can be safely isolated at home. Age, BMI, underlying hypertension, date after symptom onset, tachycardia, and systolic blood pressure at triage might relate to later hospitalization.”